# Complex networks description of the ionosphere

Shikun Lu[1,2], Hao Zhang[1], Xihai Li[2], Yihong Li[2], Chao Niu[2], Xiaoyun Yang[2], and Daizhi Liu[2]

[1]Department of Electronic and Engineering, Tsinghua University, Beijing, China.
[2]Xi'an Research Institute of Hi-Tech, Xi'an, China.

*Correspondence to:* Hao Zhang (haozhang@mail.tsinghua.edu.cn)

**Abstract.** Complex networks have emerged as an essential approach of geoscience to generate novel insights into nature of geophysical systems. To investigate the dynamic processes in the ionosphere, a directed complex network is constructed based on the probabilistic graph by the Vertical Total Electron Content (VTEC) in 2012. The results of the power-law hypothesis testing show that both the out-degree and in-degree distribution of the ionospheric network are not scale-free. Thus, the distribution of the interactions in the ionosphere is homogenous. None of the geospatial positions plays an eminently important role in the propagation of the dynamic ionospheric processes. The spatial analysis of the ionospheric network shows that the inter-connections principally exist between the neighbors in geographical space, indicating that the propagation of the dynamic processes primarily depends on the geospatial distance in the ionosphere. Moreover, the joint distribution of the edge distances shows that the dynamic processes travel further along the longitude than along the latitude. The analysis of small-world-ness indicates that the ionospheric network possesses the small-world property, which can make the ionosphere stable and efficient in the propagation of the dynamic processes. The fractal analysis shows that the ionospheric network is not self-similar in the current temporal and spatial resolution, indicating the complexity of the spatial variation for a long time in the ionosphere.

## 1 Introduction

Including large numbers of irregularities with different sizes and affected by various factors (like, solar irradiation, geomagnetic field, gravity wave and tidal wave (Kelly, 2009)), the ionosphere performs as a complex system about the spatial and temporal variation. Complex network is an efficient tool to study the characteristics of complex systems, containing a large number of interacting parts. Its application spans in various scientific fields (Zerenner et al., 2014), such as Biology (e.g., protein interaction networks), Information Technology (e.g., World Wide Web) and Social Sciences (e.g., social networks (Wang et al., 2016a, b)). The application of complex network theory to ionosphere science is still a young field, since few researches reported. The network theory was discussed by Podolská K. et al. by two abstracts in the 2010 and 2012 EGU General Assembly Conference (Podolská et al., 2010, 2012). The prior abstract wanted to examine the influence of geomagnetic disturbances and

solar activity on thermal plasma parameters. The other one attempted to find out time shifts between fundamental ionospheric parameters. Therefore, none of them tried to describe the global ionosphere based on the complex network.

In modern statistical mechanics of geophysics, especially seismological science, the idea of complex networks is receiving significant attention. Baiesi and Paczuski (2005) constructed directed networks of earthquakes by placing a link between pairs of events that were strongly correlated. Their results showed that the network was scale-free and highly clustered. Abe and Suzuki (2006) constructed growing random networks by adding an edge between two successive earthquakes and found that these earthquake networks were scale-free and small-world. The constructions of the above two networks were based on the expert judgment to add an edge and ignored the uncertainty in the system. Jiménez et al. (2008) divided the Southern California region into cells of $0.1°$ and calculated the correlation of activities in them to create networks, which showed the small-world features. Suteanu (2014) proposed a network-based method for the assessment of earthquakes' relationships in space-time-magnitude patterns and further applied the results for the study of temporal changes in volcanic seismicity patterns. Those two networks were built based on correlation, which was a linear measurement of the interactions in the objective system.

Another geophysical application of complex networks is on climate science (Nocke et al., 2015). Peron et al. (2014) also built the temperature network by correlation and regarded the global grid points as nodes. They showed that the network characteristics of the North American region marked the differences between the eastern and western regions. Such differences can be viewed as a reflection of the presence of a large network community on the west side of the continent. To depict the nonlinearity and uncertainty in the climate, information theory is introduced to construct the complex network of climate. Donges et al. (2009a, b) used complex networks to uncover a backbone structure carrying matter and energy in the global surface air temperature field. They used mutual information (MI) to construct the network which was undirected, because the mutual information was symmetric to measure the dynamical similarity of surface air temperature between regions. Hlinka et al. (2013) investigated the reliability of directed climate networks being built by conditional mutual information (CMI), using the dimensionality-reduced surface air temperature data. Compared with MI, CMI is asymmetric and able to build directed networks for the global surface air temperature. However, both MI and CMI are standard bivariate methods, which only describe the interactions between two spatial points without considering the influences of the others. So is the correlation. Probabilistic graph is an efficient method to describe the nonlinear interactions within the system from the holistic perspective (Koller and Friedman, 2009). Furthermore, similar to the seismology and climate science, the ionosphere is also distributed geographically. It is often concerned with spatial interactions and flows. These researches propose a possibility that approaches from the perspective of complex networks may also shed new light on ionospheric features. In this article, the probabilistic graph is employed to model the dynamic processes within the ionosphere and build the ionospheric complex network.

Within the global ionosphere, there are interactions among the variations over different positions. Moreover, variations over one position may cause variations over other positions. The motivation of the current study is to explore the causal interactions between the VTEC over different GIM cells within the global ionosphere based on the directed complex network. Hence, we can have a deep understanding of the dynamic processes within the ionosphere. Meanwhile, based on the causal relationship in the ionosphere, we can make a more precise prediction of the VTEC utilizing the observations obtained at the connected GIM cells in the network. Accurate prediction of the VTEC is valuable to improve the performance of GPS and ionospheric

radio propagation. We interpret the dynamic ionospheric processes as the information flow in the directed network and explore the ionospheric characteristics on a global scale. The VTEC dataset supplied by the Centre for Orbit Determination in Europe (CODE) in 2012 is selected.

The article is organized as follows. The data and method description are provided in Section 2. Furthermore, the results about the patterns of the ionospheric interactions are presented in Section 3. The scale-free topology of the ionospheric network is checked by conducting power-law hypothesis testing. The distribution of the edge distances is calculated to analyze the propagation of the dynamic processes in the ionosphere. The small-world structure of the ionospheric network is explored to examine the stability of the ionosphere. The self-similar structure in the ionosphere is investigated through fractal analysis. Section 4 discusses the summaries and conclusions.

## 2 Description of Data and Methods

### 2.1 VTEC Data Source

As a critical physical quantity of the ionosphere, VTEC carries abundant information about the variations of the ionosphere (Ercha et al., 2015). The International Global Navigation Satellite System Service (IGS) supplies global VTEC data with 2-hours' time resolution. The dataset is determined from more than 200 IGS stations within a global scale (Wei et al., 2009). CODE, as one of the analysis centers of IGS, has estimated VTEC from the dual-frequency code and phase data of GPS since April 1998 (Guo et al., 2015). In the current research, VTEC data is derived from CODE (ftp://ftp.unibe.ch/aiub/CODE) in the form of Global Ionospheric Map (GIM). The GIM ranges from $-180°$ to $180°$ along the longitude and from $-87.5°$ to $87.5°$ along the latitude. The negative values stand for the south latitude and west longitude. The size of an elementary GIM cell is $5°$ along the longitude and $2.5°$ along the latitude. Each GIM cell is defined as a variable, which is a node in the ionospheric network. The VTEC data over the GIM cell is its observation. For the decrease of the computation by reducing the variables' quantity, the size of the GIM cells has been doubled. So the latitude and longitude sizes of GIM cells become $5°$ and $10°$. The number of variables (GIM cells) is $36 \times 36$, which is 1296, because $180°$ and $-180°$ are the same for longitude. In this paper, we select the data in 2012.

### 2.2 Mapping the data to a complex network

As a complex system, the ionosphere is usually characterized by the presence of multiple interrelated aspects, which are spatially distributed. Affected by various factors, the ionosphere also involves a significant amount of uncertainty. Moreover, our observations are always noisy; even those observed aspects are often measured with some error. Thus, probability needs to be used to represent such random property. Furthermore, the probabilistic graph can efficiently describe the nonlinearity within the system from the holistic perspective (Koller and Friedman, 2009). As a result, the probabilistic graph is selected to model the interrelation and uncertainty in the ionosphere. We describe the GIM data as the realization of a multivariate probabilistic graph on the global spatial grid.

Probabilistic graphs use a graph-based representation as the basis for compactly encoding a complex probabilistic distribution over a high-dimensional space (Koller and Friedman, 2009). The probabilistic graph is a useful way of visualizing interactions between multiple variables. Therefore, in addition to inference, probabilistic graphs can also be used to discover the knowledge within the dataset. As a kind of complex networks, probabilistic graphs are constructed to represent a joint distribution by making conditional independence (CI) assumptions. The nodes in the networks represent variables, and the edges represent CI assumptions (Murphy, 2012). The absence of an edge between two nodes implies that the corresponding variables are conditionally independent given all other nodes. Based on the probability theory, we say variables $X$ and $Y$ are CI iff the conditional joint distribution can be written as a product of conditional marginal:

$$X \perp Y|Z \Longleftrightarrow p(X,Y|Z) = p(X|Z)p(Y|Z) \tag{1}$$

In our backgrounds, $X$ and $Y$ are the two given GIM cells and $Z$ represents the GIM cells except $X$ and $Y$. Thus, the analysis is performed from the holistic perspective. As suggested in the Ref. (Zerenner et al., 2014), directed complex network can offer additional knowledge, like the distinction between child and parent nodes. Thus, we construct the ionospheric networks that only include directed edges between GIM cells. Suppose two GIM cells are not directly connected (conditional independent) within the ionospheric network, there should be no interactions between these cells after eliminating all of the existing edges. The directed edges here represent the causal interactions. In other words, after the variations of VTEC over a certain GIM cell, there are some related variations appearing over other GIM cells. As following, the construction of the directed ionospheric network (also known as Bayesian probabilistic graph or Bayesian network) is introduced to describe the dynamic processes in the global ionosphere. Dynamic processes are constituted by a series of causal interactions among the GIM cells. Conditional independence tests involving sets of variables can be used to determine the existence and direction of edges (Ebert-Uphoff and Deng, 2012).

The cells in the GIMs are defined as the variables distributed throughout the globe. As the nodes on the network, the variables are separated by their own geospatial locations. The VTEC of each variable are arranged in the form of a time series with the 2-hours' time resolution. Thus, for the year 2012, the length of the observations is 4392 ($12/day \times 366day$). We employ structure learning algorithm for Bayesian network as a basis for the construction of the ionospheric networks. In our background, the measurements of the 1296 variables are all continuous. To build the directed network, we should determine the existence and directions of edges between any two variables from the holistic perspective instead of just considering the two ones. The Fast Greedy Equivalence Search (FGS) algorithm developed by Joseph Ramsey et al. (Ramsey et al., 2017) works well for large numbers of continuous variables to build Bayesian networks. This algorithm utilizes the strategy that, edges are iteratively added starting with an empty network according to maximal increases in BIC score (Schwarz, 1978). Here, the variables' distributions are assumed to be Gaussian. We use the implementation of the FGS algorithm in the TETRAD package (Version 5.3.0-2, available at http://www.phil.cmu.edu/projects/tetrad/) and make the penalty discount is 10. TETRAD possesses a convenient user interface to enter preknowledge. As the ionospheric network includes 1296 nodes and 10,985 directed edges in the globe, it is hard to fully present such a complex network. Here, we exhibit part of the ionospheric network. The result is shown in Figure 1.

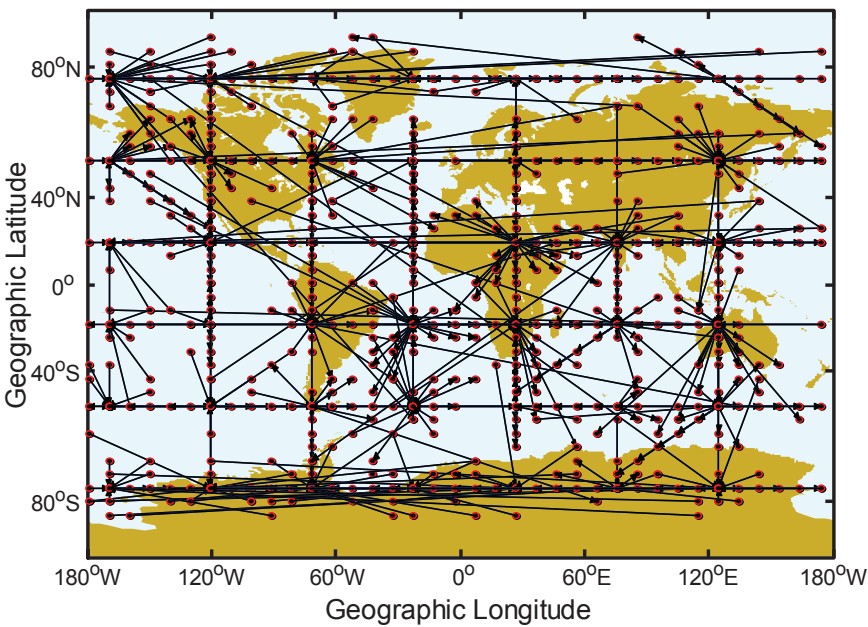

**Figure 1.** The directed complex network of the ionosphere (part). The network is developed from the VTEC dataset by the FGS algorithm. The nodes indicate the GIM cells, while the directed edges represent causal interactions between cells.

## 3 Results and Discussion

### 3.1 Degree distribution of the ionospheric network

To explore the influence of the VTEC's variation over a certain GIM cell, the degree of complex network is employed. As one of the most critical parameters to depict the nodes in a complex network, the degree is the number of edges the node possesses. Concerning ionospheric networks, the degree of a cell can be selected to quantify how many GIM cells display a causal interaction with that given cell in the globe. In other words, cells with large degree can influence large numbers of GIM cells. In the complex network, hubs refer to the nodes with large numbers of links that significantly exceed the average. Hubs have a significant effect on the system, which is described by the network. The emergence of hubs results from the scale-free property of networks (Barabási and Albert, 1999). Hence, to study the hub positions where the dynamic ionospheric processes mainly originate or converge, we have to check the scale-free topology about the degree distribution of the ionospheric network. The degree distribution is the probability distribution of these degrees over the whole network. For the directed ionospheric network, the degree distribution is divided into two different kinds, the out-degree distribution (the distribution of outgoing edges) and the in-degree distribution (the distribution of incoming edges). The degree distributions of the ionospheric network are shown in Figure 2.

It has been reported that real complex networks often exhibit scale-free properties (Barabási and Albert, 1999). This means their degree distribution follows a power law, at least asymptotically. That is, the number of links of a given node exhibits

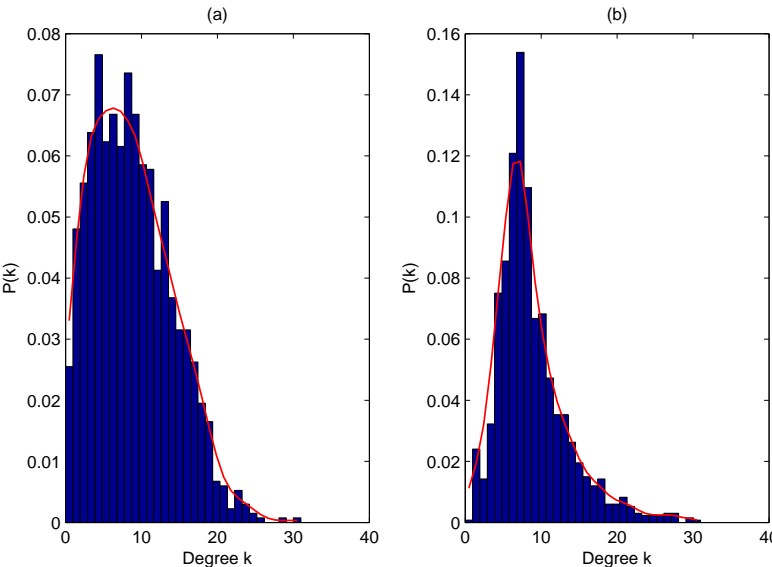

**Figure 2.** The degree distributions of the ionospheric network. (a) is the out-degree distribution; (b) is the in-degree distribution. The red curves delineate the distribution fitting.

a power law distribution $P(k) \sim k^{-\gamma}$, where $k$ is the number of links. $P(k)$ can be calculated by the statistical frequency and $\gamma$ is a parameter whose value is typically in the range, $2 < \gamma < 3$. From the distributions shown in Figure 2, it is hard to determine whether the observed degree is drawn from a power-law distribution or not. Clauset et al. (2009) presented a principled statistical framework for discerning power-law behavior in empirical data. As the method shown in the Ref.
(Clauset et al., 2009), we have tested the power-law hypothesis quantitatively. Both the results of the out-degree and in-degree distribution reject the hypothesis, indicating that the ionospheric network is not scale-free. Thus, most GIM cells approximately have the same number of edges, indicating the network of the global ionosphere is homogeneous. For the dynamic processes in the ionosphere, there is no unique spatial position acting as the sources or sinks. This property is completely different from that of the geomagnetic field. In other words, there are no visible 'hub' GIM cells for the ionospheric variations. Moreover,
from the curves of distribution fitting shown in Figure 2, we can find that both the distributions are more likely Poisson, just like the network of climate (Tsonis et al., 2007).

### 3.2 Joint distribution of the edge distances

The propagation of the dynamic processes is related to the transmission of energy or particles in the ionosphere. To analyze such transport property, the distribution of the edge distances is calculated. The edge distance is defined by the geographical distance
between the origin and destination of an edge. The height of the VTEC supplied by CODE is $H = 450km$. As the measurements are on the earth which can be regarded as a sphere, the distances between any two positions can be calculated by the arc lengths

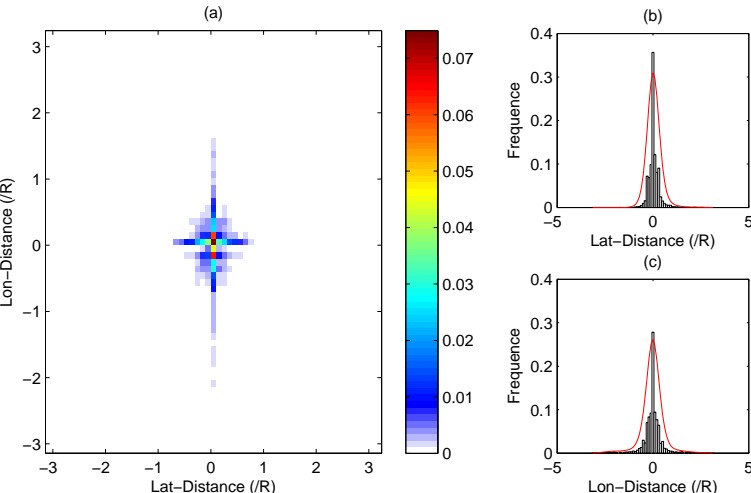

**Figure 3.** The joint distribution of the directed edge distances in the global ionospheric network. (a) is the distribution of edges against the latitudinal and longitudinal distances; (b) is the distribution of edges against their latitudinal distances; (c) is the distribution of edges against their longitudinal distances. The red curves delineate the distribution fitting.

on the sphere $d = R\theta$, where $R = R_0 + H$, $R_0$ is the earth radius and $\theta$ is the corresponding central angle. Compared with the undirected probabilistic graphs, the directed ones can provide additional knowledge about the causal interactions within the ionosphere. To study the directional characteristics of the propagation of the dynamic ionospheric processes, the edge distances are mapped to the latitude and longitude directions.

The latitudinal distances are calculated, by $d_{lat} = (lat_2 - lat_1)R$, where $lat_1$ and $lat_2$ are the latitudes of the origin and destination of the given edge. Meanwhile, the longitudinal distances are calculated, by $d_{lon} = (lon_2 - lon_1)R'$, where $lon_1$ and $lon_2$ are the longitudes of the origin and destination of the given edge. As the radii of different latitudinal circles are different, the radius of an equivalent latitudinal circle is calculated by the average of the radii of the two latitude circles where the origin and destination of the given edge are located on, i.e., $R' = \frac{1}{2}[\cos(lat_1) + \cos(lat_2)]R$. The positive signs of the

distances represent the directions of edges and can be either eastward or northward. The result is shown in Figure 3.

     As is shown in Figure 3 (a), the edges are mainly distributed around the origin of the coordinate system. Thus, the GIM cells are mostly connected with their spatial neighbors. The local connections indicate that, in the ionosphere, the propagation of the dynamic processes is primarily affected by the geospatial distance and almost satisfies the proximity principle in space. Furthermore, from the approximate symmetry along the Xlabel in Figure 3 (b) and (c), we can discover that it is almost

15 the same for the westward and eastward propagation of the dynamic processes, also for the southward and northward. From Figure 3 (b) and (c), we can find that the number of edges decreases as the increase of the absolute value of latitudinal and longitudinal distance. This phenomenon also reveals that the local interactions account for a considerable proportion in the ionospheric network. The proximal propagation may be due to the diffusion effects of charged particles in the ionosphere.

Besides, comparing the standard deviations of the edges' longitudinal and latitudinal distances, which are 0.53 and 0.28, the results show that the distribution curve along the latitude direction is steeper than that along longitude one. Therefore, the rate of decrease along the latitude is larger than that along the longitude. Accordingly, the dynamic processes are propagated more efficiently along the longitude than along the latitude. Such phenomenon may relate to the north-south currents or geomagnetic field in the ionosphere. Moreover, the ionospheric network is not entirely connected locally. Long-range edges emerge both along the latitude and longitude. The long-range propagation may be caused by the geomagnetic field or other global factors. Thus, the ionospheric network possesses a primarily ordered structure with some exceptional long-range connections.

## 3.3 Small-world structure of the ionospheric network

The small-world structure can make the system be more stable to react to the abrupt variations (Tsonis et al., 2007). Therefore, we explore the small-world structure of the ionospheric network to examine the stability of the ionosphere which is regarded as a dynamical system. Lying between the completely random and completely regular network, the small-world network is a type of graph in which any given node is likely to reach every other node by a small number of steps compared with the total number of network nodes (Gallos et al., 2007). The 'six degrees of separation' in social networks is one of the most famous examples. Watts and Strogatz (1998) found that some networks can be highly clustered, like regular lattices, yet have small characteristic path lengths, like random graphs. To investigate the small-world structure of the ionospheric network, the original network has to be reduced to an undirected graph (Abe and Suzuki, 2006, 2009). Furthermore, to mathematically describe the small-world property, two critical parameters are often selected, which are the average clustering coefficient $C$ and the average shortest path length $L$. Their definitions are shown in Equation (2), (3) and (4).

$$C_i = \frac{2\Delta_i}{k_i(k_i - 1)}, \tag{2}$$

$$C = \frac{1}{N}\sum_{i=1}^{N} C_i, \tag{3}$$

$$L = \frac{2}{N(N-1)}\sum_{i \geq j} d_{ij}. \tag{4}$$

Here, $C_i$ is the local clustering coefficient of node $i$. $k_i$ is the degree of node $i$ and $\Delta_i$ denotes the number of edges between the neighbors of node $i$ with node $i$ itself being excluded. The global clustering coefficient $C$ is defined as the average of all local clustering coefficients $C_i$. $N$ is the number of nodes and $d_{ij}$ denotes the length of the shortest path between the nodes $i$ and $j$. $d_{ij}$ is calculated by Dijkstra algorithm (Newman, 2010). Thus, $C$ describes the local connections in the ionospheric networks, while $L$ characterizes a network's connectivity structure globally (Zerenner et al., 2014).

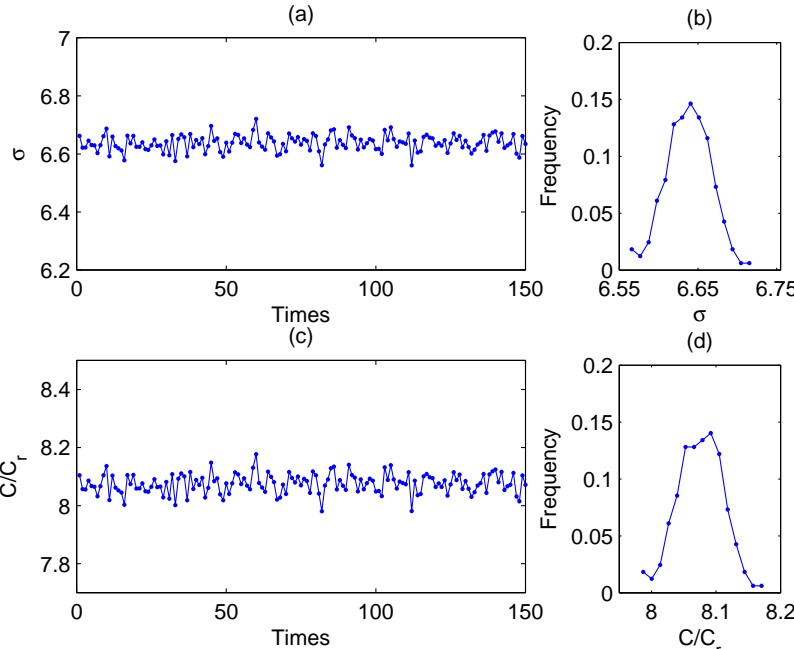

**Figure 4.** The test of the small-world structure in the ionospheric network. (a) shows the 150 results of $\sigma$; (b) shows the frequency of the results of $\sigma$; (c) shows the 150 results of $C/C_r$; (d) is the frequency of the results of $C/C_r$.

To quantitatively define a small-world network, values for the network properties must be compared with those values acquired from the equivalent random networks, which have the same degree with the given network on average. A measurement of 'small-world-ness' is proposed as follows (Humphries and Gurney, 2008; Humphries et al., 2011):

$$\sigma = \frac{C/C_r}{L/L_r}. \tag{5}$$

Here, $C$ and $L$ are the average clustering coefficient and the average shortest path length of the given network, while $C_r$ and $L_r$ are those of the equivalent random network. If the given network fulfills the conditions, $\sigma > 1$ and $C/C_r > 1$, it meets the small-world criteria. To reduce the impact of randomness, the results shown in Figure 4 are calculated by 150 random networks.

From Figure 4 (a) and (c), we can find that the results all satisfy $\sigma > 1$ and $C/C_r > 1$. Shown in Figure 4 (b) and (d), the frequencies are approximately Gaussian, and the standard deviations are 0.028 and 0.035. Such small standard deviations indicate the results are close to the real values (the averages) 6.64 and 8.08. Therefore, the ionospheric network behaves as a small-world graph. The propagation of the dynamic processes in the ionosphere presents small-world property. Just as the small-world property in the atmosphere (Donges et al., 2009b), such ionospheric property also results from the teleconnections beyond the geospatial distance in the ionospheric network. Such teleconnections play an important role in stabilizing the ionosphere system and make the dynamic ionospheric processes transferred efficiently (Donges et al., 2009b; Tsonis et al.,

2007). If a disturbance is generated somewhere in the ionosphere, the small-world structure of the ionospheric network allows the ionosphere to react quickly and coherently to those variations. This propagation mechanism of the dynamic process can diffuse local variations thereby reducing the possibility of prolonged local anomalies and providing more stability for the global ionosphere. Thus, chances of major ionospheric shifts are reduced.

## 3.4 Fractal nature within the ionospheric network

The spatial prediction, especially regional prediction, depends heavily on the self-similarity in the ionosphere. Thus, we investigate the self-similar structure in the ionosphere through fractal analysis, which can also provide a deeper understanding of the ionospheric network. Song et al. (2005) applied the Box-covering algorithm to demonstrate the existence of fractality in many real networks. By the regular fractal analysis, the fractal dimension can be calculated through the Box-covering algorithm. The relation between the boxes' minimum number ($N_B(l_B)$) needed to cover the network and the box's size($l_B$) obeys the power law as follows:

$$N_B(l_B) \sim l_B^{-d_B}. \tag{6}$$

If Equation (6) is satisfied by the given network, then $\lg N_B(l_B) \sim -d_B \lg l_B$. Thus, the fractal dimension $d_B$ can be obtained by $d_B = -\lim_{l_B \to 0} \frac{\lg N_B(l_B)}{\lg l_B}$. In the practical application, $d_B$ is usually obtained by the negative of the slope of the line fitted by $\lg N_B(l_B)$ against $\lg l_B$ (Molontay, 2015).

The ultimate goal of Box-covering algorithms is to locate the optimum solution, i.e., to identify the $N_B(l_B)$ value for any given box size $l_B$. Song et al. (2007) demonstrated that this problem could be mapped to the graph coloring problem, which was known to belong to the family of NP-hard problems. To solve the problem efficiently, the greedy coloring algorithm is utilized. As the Ref. (Song et al., 2005) suggested, the greedy coloring algorithm can both reveal the self-similar structure and determine the fractal dimension. However, a dual network should be constructed beforehand by the application of a renormalization procedure which coarsely grains the ionospheric network into boxes containing nodes within a given size $l_B$ (Song et al., 2007). In the greedy coloring algorithm, boxes are treated as the nodes of the dual network. Node coloring is a well-known procedure, where colors are assigned to each node of a network. Accordingly, the minimum number of boxes $N_B(l_B)$ is equal to the minimum required number of colors. The details of the algorithm are shown in the Ref. (Song et al., 2007). Because the results of the greedy coloring algorithm depend on the original coloring sequence, we randomly reshuffle the original coloring sequence and apply the greedy coloring algorithm for 10,000 times to investigate the quality of the algorithm. The results are shown in Figure 5.

Here, $l_B$ ranges from 1 to $l_B^{max}$, which is the maximum distance in the ionospheric network plus 1. Based on the Box-covering algorithm, the distance $l$ between the GIM cells in the same box should fulfill $l < l_B$. Therefore, when $l_B$ is 1 and $l_B^{max}$, the minimum number of boxes $N_B$ is the number of GIM cells and 1, as is shown in Figure 5 (b) and (c) where $l_B^{max} = 8$. Shown in Figure 5 (a), the distribution curves for all box sizes $l_B$ are narrow Gaussian distributions, indicating that almost every implementation of the algorithm yields a solution close to the optimal. Accordingly, we can use the mean to stand for the optimal value when we use Least Square fitting to measure the fractal dimension $d_B$. Figure 5 (b) is log-log plot, and (c)

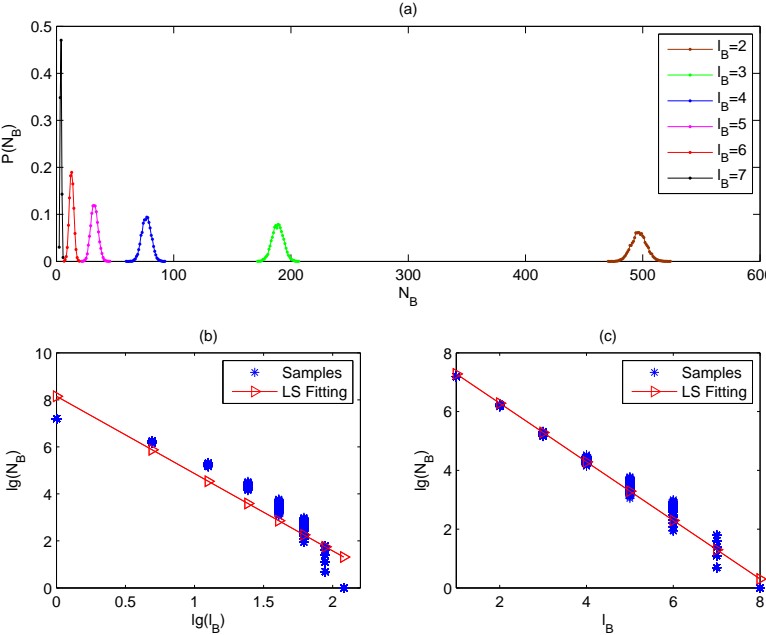

**Figure 5.** The results of 10,000 times of greedy coloring algorithm implementation for the ionospheric network. (a) shows the probability distribution $P(N_B)$ of the minimum box number of $N_B$ against different box sizes $l_B$; (b) shows the 10,000 results of $N_B$ against different $l_B$ on the log-log scale and the corresponding Least Square fitting; (c) shows the 10,000 results of $\lg(N_B)$ against different $l_B$ and the corresponding Least Square fitting.

is semi-log plot for Y-axis. Comparing those two subfigures, it is obvious that the relationship between $N(l_B)$ and $l_B$ follows $N(l_B) \sim 10^{-l_B}$ rather than $N(l_B) \sim l_B^{-d_B}$.

Therefore, in the current temporal and spatial resolution, the ionospheric network does not possess self-similar structure, indicating the complexity of the ionospheric spatial variation for a long time. Construction of the long-term geospatial model of the ionosphere is still a challenging work. We believe that because of the non-fractal property, the predictability of the ionosphere for one year should decrease. Spatial characteristics within the ionosphere differ complexly and dramatically with the variation of regions for one year. Such complex spatial variations in the current resolution may disrupt the similarity in the ionosphere. To further investigate the self-similarity within the ionosphere, the time window and space resolution of the ionospheric observations should be considered. In our background, the self-similarity in the ionosphere may be detected by the observations of high temporal and spatial resolution.

## 4 Conclusions

The ionosphere can be regarded as a spatially extended complex system. Therefore, the complex network is used to analyze the dynamic processes in the global ionosphere based on the VTEC from IGS. As a Bayesian probabilistic graph, the ionospheric network is constructed based on the conditional independence theory by FGS algorithm. The edges of the network represent the causal relationships between any two GIM cells from the holistic perspective. We have analyzed the structure of the directed ionospheric network. The results of the power-law hypothesis test show that both the out-degree and in-degree distribution of the ionospheric network are not scale-free. The ionospheric network is homogenous. None of the geospatial positions plays an eminently important role in the propagation of dynamic ionospheric processes. Based on the latitudinal and longitudinal distances between the beginnings and ends of the edges, the joint distribution is analyzed to explore the propagation of the dynamic processes in the ionosphere. The results show the edges principally exist between the neighbors in geographic space, indicating that the propagation of the dynamic processes mainly atisfies the proximity principle in the ionosphere. Moreover, the joint distribution of the edge distances shows that the dynamic processes travel more efficiently along the longitude than along the latitude. Also, the small-world structure is studied to examine the stability of the ionosphere. The small-world-ness is found to be larger than 1. Meanwhile, the clustering coefficient is larger than those of the classical random networks. Thus, the ionospheric network possesses small-world property, which makes the ionosphere stable and efficient in the propagation of the dynamic processes. Also, the analysis of the self-similar structure shows the ionospheric network is not fractal in the current resolution, indicating the complexity of the spatial variation for a long time in the ionosphere. In general, the complex network provides a peculiar perspective on the ionosphere research. Depending on the choice of nodes, edges and methods, ionospheric networks may take different forms to study different properties of the ionosphere.

*Code availability.* Code are available by email request.

*Data availability.* VTEC data is derived from CODE (ftp://ftp.unibe.ch/aiub/CODE) in the form of Global Ionospheric Map.

*Competing interests.* No competing interests are present

*Acknowledgements.* This work was supported by the National Natural Science Foundation of China (41374154). We are grateful to Adam Woods from CIRES, University of Colorado, David Skaggs Research Center, Rolf Dach and Stefan Schaer from Astronomical Institute, University of Bern. They are all so kind to help us on the data obtaining.

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
