# Peer review of "Complex networks description of the ionosphere"

_Nonlinear Processes in Geophysics, 2017_

## Short Comment (SC1) · 13 Aug 2017

1 For line 5 in Page 8, the expression $\log N_B(l_B) \sim -d_B \log l_B$ should be corrected as $\lg N_B(l_B) \sim -d_B \lg l_B$.

2 the world 'which ' should be deleted in line 24 in Page 7.

3 the world 'the ' should be deleted in line 18 in Page 7.

4 'mathmatical definition of infromation flow' should be corrected as 'mathematical definition of information flow'in line 1 in page 4.

---

## Author Comment (AC1) · 13 Aug 2017

Thanks for the suggestions.

The corrections have been made in the revised manuscript.

---

## Short Comment (SC2) · 19 Aug 2017

This paper have provided new insights on the global ionosphere by the application of complex network and make description of the ionospheric network to learn more about the variations in the ionosphere. But I still have several suggestions as follows,

1 Figure 2 shows the degree distributions of the network of ionosphere. To show the distribution intuitively, I suggest the curve of distribution fitting to be added in the figure.

2 Subsection 3.3 investigated the small-world structure. For the small world structure caused by the long-range edges, more explanation may be needed.

3 After careful reading the paper, I think some statements in the article need to be improved, like the subtitle of 2.2 and 3.2. So, I suggest you read more paper about the application of complex network and make an overall revision about the current

manuscript.

I hope these suggestions can help to improve the manuscript.

---

## Author Comment (AC2) · 20 Aug 2017

Thanks very much for the positive comments and constructive suggestions, which are helpful to improve the quality of the manuscript.

1 Thanks for the helpful advice. The curve of distribution fitting by Kernel smoothing method has been added in the Figure (as is shown in the following) to show the distribution intuitively.

2 As introduced in subsection 3.3, the small-world structure is caused by the long-range edges. The long-range edges make the complex network stable and efficient in information transmission. For example, a disturbance is generated somewhere in the complex network. However, the small world structure of the network allows the system to respond quickly and coherently to variations introduced into it. This mechanism about information transmission diffuses local variations thereby reducing the possibility of prolonged local anomalies and providing more stability for the system. Thus, chances of major shifts are reduced.

3 Thanks for the advice. After comprehensive thinking, the subtitle of 2.2 and 3.2 have been replaced with "Mapping the data to a complex network" and "Joint distribution of the edge spans". We will learn more about the application about the complex and make an overall revision which will be shown in the revised manuscript.
* * *
[Figure]

[Figure]

**Fig. 1.**

---

## Referee Comment (RC1) · Anonymous Referee #1 · 14 Nov 2017

In recent decades, attentions on complex networks have been more and more paid to the field of geoscience as a powerful tool in investigations. Particularly, in the study of climatology and seismology. In this paper, the authors firstly introduce this method to construct a directed complex network to investigate the information flow in ionosphere. Some new results are gained that both the out-degree and in-degree distribution of the ionospheric network are not scale-free. The topological structure of the ionospheric information network is homogeneous. The spatial variation of the ionospheric network shows the connection principally exist between the neighbors in space, indicating that in ionosphere the information transmission is mainly based on the spatial distance. Since this is the first time that ionospheric data are used to construct such a network, the results are helpful in understanding some special characteristics of the ionosphere. The followings are a few suggestions to the authors as a reference: 1, Can authors make a simple comparison of results of this paper with other relatively similar earlier

published networks like surface temperate data structed networks. Such comparison may provide some useful hints for the further development of the complex network construction 2, also, is it possible to explain the results of networks in the real ionospheric features 3, the data used in this paper only with one year's time span, and time resolution is two hours and ranges from $-180$âŮę to $180$âŮę along the longitude and from $-87.5$âŮę to $87.5$âŮę along the latitude with a revolution of 10 and 5 degree in longitude and latitude. Is the revolution affects the results? For example, in such a revolution, ionospheric equatorial anomaly, small-scaled irregularities are excluded, then how can we say the ionospheric network is not fractal?

---

## Referee Comment (RC2) · Anonymous Referee #2 · 14 Nov 2017

Review for manuscript "Complex networks description of ionosphere" by Shikun Lu, Hao Zhang, Xihai Li, Yihong Li, Chao Niu, Xiaoyun Yang, and Daizhi Liu

Content of the study: The authors study spatial connectivity patterns of global Vertical Total Electron Content(VTEC) in the ionosphere using Bayesian Networks. The key results are that the network representing conditional dependencies of the VTEC 1) is not scale, 2) is small-world and 3) is not fractal.

Overall evaluation: This study applies existing methods and concepts to observational gridded data (VTEC). The technical analyses seem to thorough. However, since there is thus no methodological advance, I would expect some physical / mechanistic motivation for the conducted research, as well as physical interpretations of the results: Why is the network not scale-free? Why was this tested? What are the implications? Similarly, what might be the reason for the small-worldness? And what are the implications of the network not being fractal? What can we learn from the analysis about the physical complex system under study? There are only very vague statements addressing these questions and therefore, in its present state, it seems a bit like the network was constructed and typical network characteristics were determined simply because it's possible to do so, and not because of a driving scientific hypothesis. In other words, the results are original, but it is not clear why they are meaningful. I do not recommend rejection of the paper, because the analyses and results seem to be correct, but leave it to the editor to decide to what degree interpretations and discussions of the results are expected for the journal.

Comments: 1. The manuscript should be proof-read by a native English speaker. 2. In its current presentation, the paper is hardly reproducible, because it is not clear how, specifically, the Bayesian Network was constructed. I strongly suggest to add a paragraph where this is explained in detail. 3. I'm not sure what I should learn from Fig.1 4. Regarding the spatial variations: The distance between connected grid cells is measured in terms of degrees (lat/lon). However, the spatial distance (in units of meters) between meridians varies with latitude (they are weighted with cos(lat)), and this severely biases the results shown in Fig.3: At high latitudes, a distance of, say, one degree, corresponds to a much shorter distance in space than at low latitudes, which produces apparent long-ranged connections if only measured in degrees. Before drawing any conclusions from the apparent asymmetry between latitudinal and longitudinal information transport, the distances should be translated to actual spatial distances, measured in meters!

---

## Author Comment (AC3) · 15 Dec 2017

The authors would like to express our sincere gratitude for all the constructive comments on our manuscript. The comments and suggestions are very helpful for the improvement of our paper. In what follows, we present detailed comments in response to the individual points raised by the reviewer and elaborate on how the manuscript has been revised.

**(The line numbers referred in the response are those in the manuscript tracking changes. This manuscript is provided as the supplement.)**

**1   General Comments**

**Comments:** *In recent decades, attentions on complex networks have been more and more paid to the field of geoscience as a powerful tool in investigations, particularly, in the study of climatology and seismology. In this paper, the authors firstly introduce this method to construct a directed complex network to investigate the information flow in the ionosphere. Some new results are gained that both the out-degree and in-degree distribution of the ionospheric network are not scale-free. The topological structure of the ionospheric information network is homogeneous. The spatial variation of the ionospheric network shows the connections principally exist between the neighbors in space, indicating that in the ionosphere the information transmission is mainly based on the spatial distance. Since this is the first time that ionospheric data are used to construct such a network, the results are helpful in understanding some special characteristics of the ionosphere.*

**Response:** We thank the reviewer very much for these positive comments and present the details of responses to the concerns in the following part. In the document that follows, we describe the associated modifications made to the original version of the paper and address the comments of the reviewers separately.

**2   Major Comments**

**Issue 1:** *Can authors make a simple comparison of results of this paper with other relatively similar earlier published networks like surface temperate data structed networks. Such comparison may provide some useful hints for the further development of the complex network construction.*

**Response 1:** Many thanks for the comment. As the reviewer suggested, we have made a comparison between the network of this paper with other relatively similar

earlier published networks. The details are shown as follows,

Peron et al. (2014) built the temperature network by correlation and regarded the global grid points as nodes. They showed that the network characteristics of the North American region have marked the differences between the eastern and western regions. Such differences were a reflection of the presence of a large network community on the western side of the continent. Correlation is a linear measurement of the dynamics in the climate system. To depict the nonlinearity and uncertainty in the climate, information theory is introduced to construct the complex network. Donges et al. (2009a,b) used complex networks to uncover a backbone structure carrying matter and energy in the global surface air temperature field. They used mutual information (MI) to construct the network which was undirected, because the mutual information is symmetric to measure the dynamical similarity of surface air temperature between regions. Hlinka et. al (2013) investigated the reliability of directed climate networks being constructed by conditional mutual information (CMI), using the dimensionality-reduced surface air temperature data. Compared with MI, CMI is asymmetric and able to build directed networks for the global surface air temperature. However, both MI and CMI are standard bivariate methods, which only describe the interactions between two spatial points without considering the influences of the others. So is the correlation. The revisions are shown in lines [15-27] of page 2.

**Issue 2:** *Is it possible to explain the results of networks in the real ionospheric features*

**Response 2:** Many thanks for the comment. We try to explain the results of networks in the real ionospheric features, but the explanations still need to be further verified by observations.

**1** To explore the influence of the VTEC's variation over a certain GIM cell, the degree of complex network is employed. As one of the most critical parameters to depict the nodes in a complex network, the degree is the number of edges the node possesses. Concerning ionospheric networks, the degree of a cell can be selected to quantify how

many GIM cells display a causal interaction with that given cell in the globe. In other words, cells with large degree can influence large numbers of GIM cells. We have checked the scale-free topology of the ionospheric network by conducting power-law hypothesis testing about the degree distribution. The result shows that the network of the global ionosphere is not scale-free. Thus, there are no visible hub positions for the dynamic process in the ionosphere. The ionospheric network is homogenous. There are no unique positions acting as the sources or sinks of the variations within the ionosphere. This property is completely different from that of the geomagnetic field. The revisions are shown in lines [6-15] of page 5 and lines [9-15] of page 6.

**2** As the referee 2 suggested, the distance between connected grid cells is measured in terms of meters instead of degrees. The propagation of the dynamic processes is related to the transmission of energy or particles in the ionosphere. In order to analyze such transport property, the distribution of the edge distances is calculated. The edge distance is defined by the geographical distance between the origin and destination of an edge. The positive signs of the distances represent the directions of edges and can be either eastward or northward. The results show that the propagation is mainly affected by the geospatial distance and almost satisfies the proximity principle in space. Meanwhile, there are also some exceptional long-range edges existing in the ionospheric network. Accordingly, most of the dynamic processes in the ionosphere are locally propagated with some long-range propagation. Such phenomenon indicates the complexity of the inner ionospheric variations in the globe. The proximal propagation may be due to the diffusion effects of charged particles in the ionosphere, while the long-range one may be caused by the geomagnetic field or other global factors. These explanations still need to be further verified by observations. The revisions are shown in lines [2-15] of page 7 and lines [1,5-14] of page 8.

**3** We explore the small-world structure of the ionospheric network to check the stability of the ionosphere which is regarded as a dynamical system, because the small-world structure can make the system be stable to react to the abrupt variations. The results

indicate that the dynamic processes in the ionosphere present small-world property. Just as the small-world property in the atmosphere, such ionospheric property also results from the teleconnections in the ionospheric network. The teleconnections make the dynamic processes be propagated within the ionospheric network efficiently. If a disturbance is generated somewhere in the ionosphere, the small-world structure of the ionospheric network allows the ionosphere to react quickly and coherently to the variations introduced into the ionosphere. This propagation mechanism of dynamic processes can diffuse local variations, thereby reducing the possibility of prolonged local anomalies and providing more stability for the global ionosphere. Thus, chances of major ionospheric shifts are reduced. The revisions are shown in lines [16-18] of page 8 and lines [2-4] of page 10.

**4** The spatial prediction, especially regional prediction, depends heavily on the self-similarity in the ionosphere. Thus, we investigate the self-similar structure in the ionosphere through fractal analysis, which shows the ionospheric network is not self-similar in the current temporal and spatial resolution. Such phenomenon shows that because of the non-fractal property, the predictability of the ionosphere for one year should decrease. Construction of the long-term geospatial model of the ionosphere is still a challenging work. Spatial characteristics within the ionosphere differ complexly and dramatically with the variation of regions for one year. Such complex spatial variations in the current resolution may disrupt the similarity in the ionosphere. To further investigate the self-similarity within the ionosphere, the temporal and spatial resolution of the ionospheric observations should be considered. In our background, the self-similarity in the ionosphere may be detected by the observations of high temporal and spatial resolution. The revisions are shown in lines [11-13] of page 10, lines [11-14] of page 11 and lines [1-2] of page 12.

**Issue 3:** *The data used in this paper only with one year's time span, and time resolution is two hours and ranges from $-180°$ to $180°$ along the longitude and from $-87.5°$ to $87.5°$ along the latitude with a revolution of 10 and 5 degree in longitude and latitude.*

*Is the revolution affects the results? For example, in such a revolution, ionospheric equatorial anomaly, small-scaled irregularities are excluded, then how can we say the ionospheric network is not fractal?*

**Response 3:** Many thanks for the comment.

As the reviewer suggested, the fractal analysis of the ionosphere really depends on the resolution of the observations. The ionosphere is a dynamic system containing complex temporal and spatial variations. As a description of the ionosphere, the construction of the complex network is also influenced by the time window and spatial position of the ionospheric observations. As the reviewer suggested, in the current revolution, ionospheric equatorial anomaly, small-scaled irregularities are excluded. The previous statement of this conclusion is not precise. For rigorous expression, the conclusion should be presented as follows,

"Therefore, in the current temporal and spatial resolution, the ionospheric network does not have self-similar structure, indicating the complexity of the ionospheric temporal and spatial variations. To further investigate the self-similarity within the ionosphere, the time window and space resolution of the ionospheric observations should be considered. In our background, the self-similarity in the ionosphere may be detected by the observations of high temporal and spatial resolution." The revisions are shown in lines [9-10, 13-14] of page 11 and lines [1-2] of page 12.

Please also note the supplement to this comment:
https://www.nonlin-processes-geophys-discuss.net/npg-2017-29/npg-2017-29-AC3-supplement.pdf

**Supplement:**

**Complex networks description of the ionosphere**

Shikun Lu1,2, Hao Zhang1, Xihai Li2, Yihong Li2, Chao Niu2, Xiaoyun Yang2, and Daizhi Liu2 1Department of Electronic and Engineering, Tsinghua University, Beijing, China. 2Xi'an Research Institute of Hi-Tech, Xi'an, China. *Correspondence to:* Hao Zhang (haozhang@mail.tsinghua.edu.cn)

**Abstract.** Complex networks have emerged as an **essential** approach of geoscience to generate novel insights into nature of geophysical systems. To investigate the dynamic processes in the ionosphere, a directed complex network is constructed based on the **probabilistic graph** by the **Vertical Total Electron Content (VTEC)** in 2012. The results of the power-law hypothesis testing show that both the out-degree and in-degree distribution of the ionospheric network are not scale-free. Thus,

- 5 the distribution of the interactions in the ionosphere is homogenous. None of the geospatial positions plays an eminently important role in the propagation of the dynamic ionospheric processes. The spatial analysis of the ionospheric network shows that the inter-connections principally exist between the neighbors in geographical space, indicating that the propagation of the dynamic processes primarily depends on the geospatial distance in the ionosphere. Moreover, the joint distribution of the edge distances shows that the dynamic processes travel further along the longitude than along the latitude. The analysis
- 10 of small-world-ness indicates that the ionospheric network possesses the small-world property, which can make the ionosphere stable and efficient in the propagation of the dynamic processes. The fractal analysis shows that the ionospheric network is not self-similar in the current temporal and spatial resolution, indicating the complexity of the spatial variation for a long time in the ionosphere.

*Copyright statement.* All authors have approved the manuscript for submission. The content of the manuscript has not been published or submitted elsewhere.

[revised manuscript text omitted]

---

## Author Comment (AC4) · 15 Dec 2017

The authors would like to express our sincere gratitude for all the constructive comments on our manuscript. The comments and suggestions are very helpful for the improvement of our paper. In what follows, we present detailed comments in response to the individual points raised by the reviewer and elaborate on how the manuscript has been revised.

**(The line numbers referred in the response are those in the manuscript tracking changes. This manuscript is provided as the supplement.)**

**1 General Comments**

**Comments:** *This study applies existing methods and concepts to observational grid-ded data (VTEC). The technical analyses seem to be thorough. However, since there is thus no methodological advance, I would expect some physical / mechanistic motivation for the conducted research, as well as physical interpretations of the results: Why is the network not scale-free? Why was this tested? What are the implications? Similarly, what might be the reason for the small-worldness? And what are the implications of the network not being fractal? What can we learn from the analysis about the physical complex system under study? There are only very vague statements addressing these questions and therefore, in its present state, it seems a bit like the network was constructed and typical network characteristics were determined simply because it's possible to do so, and not because of a driving scientific hypothesis. In other words, the results are original, but it is not clear why they are meaningful. I do not recommend rejection of the paper, because the analyses and results seem to be correct, but leave it to the editor to decide to what degree interpretations and discussions of the results are expected for the journal.*

**Response:** We thank the reviewer very much for these positive comments. The reviewer's general comments are of great value to help us rethink the motivation, physical interpretations and applications of the results. We try to provide clear statements in the manuscript to address the questions above.

Within the global ionosphere, there are interactions among the variations over different positions. Moreover, variations over one position may cause variations over other positions. The motivation of the current study is to explore the causal interactions between the VTEC over different GIM cells within the global ionosphere based on the directed complex network. Hence, we can have a deep understanding of the dynamic processes within the ionosphere. Meanwhile, based on the causal relationship in the ionosphere, we can make a more precise prediction of the VTEC utilizing the observations obtained at the connected GIM cells in the network. Accurate prediction of the VTEC is valuable to improve the performance of GPS and ionospheric radio propagation. We interpret the dynamic ionospheric processes as the information flow in the directed network and explore the ionospheric characteristics on the global scale. The revisions are shown in lines [33-35] of page 2 and lines [1-4] of page 3.

The detailed explanations are as follows:

**1** To explore the influence of the VTEC's variation over a certain GIM cell, the degree of complex network is employed. As one of the most critical parameters to depict the nodes in a complex network, the degree is the number of edges the node possesses. Concerning ionospheric networks, the degree of a cell can be selected to quantify how many GIM cells display a causal interaction with that given cell in the globe. In other words, cells with large degree can influence large numbers of GIM cells. We have checked the scale-free topology of the ionospheric network by conducting power-law hypothesis testing about the degree distribution. The result shows that the network of the global ionosphere is not scale-free. Thus, there are no visible hub positions for the dynamic process in the ionosphere. The ionospheric network is homogenous. There are no unique positions acting as the sources or sinks of the variations within the ionosphere. This property is completely different from that of the geomagnetic field. The revisions are shown in lines [6-15] of page 5 and lines [9-15] of page 6.

**2** As the referee 2 suggested, the distance between connected grid cells is measured in terms of meters instead of degrees. The propagation of the dynamic processes is related to the transmission of energy or particles in the ionosphere. In order to analyze such transport property, the distribution of the edge distances is calculated. The edge distance is defined by the geographical distance between the origin and destination of an edge. The positive signs of the distances represent the directions of edges and can be either eastward or northward. The results show that the propagation is mainly affected by the geospatial distance and almost satisfies the proximity principle in space. Meanwhile, there are also some exceptional long-range edges existing in the ionospheric network. Accordingly, most of the dynamic processes in the ionosphere are locally propagated with some long-range propagation. Such phenomenon indicates the complexity of the inner ionospheric variations in the globe. The proximal propagation may be due to the diffusion effects of charged particles in the ionosphere, while the long-range one may be caused by the geomagnetic field or other global factors. These explanations still need to be further verified by observations. The revisions are shown in lines [2-15] of page 7 and lines [1,5-14] of page 8.

**3** We explore the small-world structure of the ionospheric network to check the stability of the ionosphere which is regarded as a dynamical system, because the small-world structure can make the system be stable to react to the abrupt variations. The results indicate that the dynamic processes in the ionosphere present small-world property. Just as the small-world property in the atmosphere, such ionospheric property also results from the teleconnections in the ionospheric network. The teleconnections make the dynamic processes be propagated within the ionospheric network efficiently. If a disturbance is generated somewhere in the ionosphere, the small-world structure of the ionospheric network allows the ionosphere to react quickly and coherently to the variations introduced into the ionosphere. This propagation mechanism of dynamic processes can diffuse local variations, thereby reducing the possibility of prolonged local anomalies and providing more stability for the global ionosphere. Thus, chances of major ionospheric shifts are reduced. The revisions are shown in lines [16-18] of page 8 and lines [2-4] of page 10.

**4** The spatial prediction, especially regional prediction, depends heavily on the self-similarity in the ionosphere. Thus, we investigate the self-similar structure in the ionosphere through fractal analysis, which shows the ionospheric network is not self-similar in the current temporal and spatial resolution. Such phenomenon shows that because of the non-fractal property, the predictability of the ionosphere for one year should decrease. Construction of the long-term geospatial model of the ionosphere is still a challenging work. Spatial characteristics within the ionosphere differ complexly and

dramatically with the variation of regions for one year. Such complex spatial variations in the current resolution may disrupt the similarity in the ionosphere. To further investigate the self-similarity within the ionosphere, the temporal and spatial resolution of the ionospheric observations should be considered. In our background, the self-similarity in the ionosphere may be detected by the observations of high temporal and spatial resolution. The revisions are shown in lines [11-13] of page 10, lines [11-14] of page 11 and lines [1-2] of page 12.

**2  Major Issues**

**Issue 1:**  *The manuscript should be proof-read by a native English speaker.*

**Response 1:** Many thanks for the comment. We are sorry about the poor writing. As the reviewer suggested, we have invited a native English speaker to proof-read the manuscript. The revisions are tracked using bold face in the manuscript tracking changes.

**Issue 2:**  *In its current presentation, the paper is hardly reproducible, because it is not clear how, specifically, the Bayesian Network was constructed. I strongly suggest to add a paragraph where this is explained in detail.*

**Response 2:** Many thanks for the comment. We are sorry for the unclear explanation. As the reviewer suggested, a paragraph is added to explain the construction of Bayesian network in detail.

The cells in the GIMs are defined as the variables distributed throughout the globe. As the nodes on the network, the variables are separated by their own geospatial locations. The VTEC measurements of each variable are arranged in the form of a time series with the 2-hours' time resolution. Thus, for the year 2012, the length of the observations is 4392 ($12/day \times 366day$). We employ structure learning algorithm for Bayesian network as a basis for the construction of the ionospheric networks. In our background, the measurements of the 1296 variables are all continuous. To build the directed network, we should determine the existence and directions of edges between any two variables from the holistic perspective instead of just considering the two ones. The Fast Greedy Equivalence Search (FGS) algorithm developed by Joseph Ramsey et al. works well for large numbers of continuous variables to build Bayesian networks. This algorithm utilizes the strategy that, edges are iteratively added starting with an empty network according to maximal increases in BIC score. Here, the variables' distributions are assumed to be Gaussian. We use the implementation of the FGS algorithm in the TETRAD package (Version 5.3.0-2, available at http://www.phil.cmu.edu/projects/tetrad/) and make the penalty discount is 10. TETRAD possesses a convenient user interface to enter preknowledge. The revisions are shown in lines [24-34] of page 4.

**Issue 3:** *I'm not sure what I should learn from Fig.1*

**Response 3:** Many thanks for the comment. We are sorry for the unclear presentation.

Fig. 1 is presented for providing some intuitive knowledge about the complex network of the ionosphere. As the ionospheric network includes 1296 nodes and 10,985 directed edges in the globe, it is hard to fully present such a complex network. Here, we exhibit part of the ionospheric network. The revisions are shown in Figure 1 and lines [1-3] of page 5.

**Issue 4:** *Regarding the spatial variations: The distance between connected grid cells is measured in terms of degrees (lat/lon). However, the spatial distance (in units of meters) between meridians varies with latitude (they are weighted with cos(lat)), and this severely biases the results shown in Fig.3: At high latitudes, a distance of, say, one degree, corresponds to a much shorter distance in space than at low latitudes, which produces apparent long-ranged connections if only measured in degrees. Before drawing any conclusions from the apparent asymmetry between latitudinal and longitudinal*

*information transport, the distances should be translated to actual spatial distances, measured in meters!*

**Response 4:** Many thanks for the comment.

We all approve the reviewer's comment that the spatial distance (in units of meters) between meridians varies with latitude. As the reviewer infers, the results have severely biases. The latitude and longitude spans has been changed to the latitude and longitude distances. The height of the VTEC measurements supplied by CODE is $H = 450km$. As the measurements are on the earth which can be regarded as a sphere, the distances between any two positions can be calculated by the arc lengths on the sphere $d = R\theta$, where $R = R_0 + H$ and $R_0$ is the earth radius, $\theta$ is the corresponding central angle. In order to study the directional characteristics of the propagation of the dynamic ionospheric processes, the distances are mapped to the latitude and longitude directions. The revisions are shown in lines [3-7] of page 7.

Please also note the supplement to this comment:
https://www.nonlin-processes-geophys-discuss.net/npg-2017-29/npg-2017-29-AC4-supplement.pdf

**Supplement:**

**Complex networks description of the ionosphere**

Shikun Lu[1,2], Hao Zhang[1], Xihai Li[2], Yihong Li[2], Chao Niu[2], Xiaoyun Yang[2], and Daizhi Liu[2]

[1]Department of Electronic and Engineering, Tsinghua University, Beijing, China.
[2]Xi'an Research Institute of Hi-Tech, Xi'an, China.

*Correspondence to:* Hao Zhang (haozhang@mail.tsinghua.edu.cn)

**Abstract.** Complex networks have emerged as an **essential** approach of geoscience to generate novel insights into nature of geophysical systems. To investigate the dynamic processes in the ionosphere, a directed complex network is constructed based on the **probabilistic graph** by the **Vertical Total Electron Content (VTEC)** in 2012. The results of the power-law hypothesis testing show that both the out-degree and in-degree distribution of the ionospheric network are not scale-free. Thus,

5 **the distribution of the interactions in the ionosphere is homogenous. None of the geospatial positions plays an eminently important role in the propagation of the dynamic ionospheric processes.** The spatial analysis of the ionospheric network shows that the inter-connections principally exist between the neighbors in geographical space, indicating that the **propagation of the dynamic processes primarily** depends on the geospatial distance in the ionosphere. Moreover, the joint distribution of the edge **distances** shows that the dynamic processes travel further along **the longitude than along the latitude**. The analysis

10 of small-world-ness indicates that the ionospheric network possesses the small-world property, which can make the ionosphere stable and efficient in the **propagation of the dynamic processes**. The fractal analysis shows that the ionospheric network is not self-similar **in the current temporal and spatial resolution**, indicating the complexity of the spatial variation for a long time in the ionosphere.

*Copyright statement.* All authors have approved the manuscript for submission. The content of the manuscript has not been published or

15 submitted elsewhere.

[revised manuscript text omitted]

---

## Author Response (AR2)

**Response to Review of Manuscript:**

**Complex networks description of the ionosphere**

**(Anonymous Referee 3)**

Shikun Lu, Hao Zhang, Xihai Li, Yihong Li, Chao Niu, Xiaoyun Yang, and Daizhi Liu

We would like to acknowledge all the reviewers for their thorough reviews and constructive comments on our manuscript. The comments and suggestions are very helpful for the improvement of our paper. Following these comments, we have conducted a more thorough and rigorous review of our work and made a comprehensive revision. We believe that the manuscript has been seriously revised according to the reviewers' comments. In the document that follows, we describe the associated modifications made to the original version of the paper and address the comments of the reviewers.

The authors would like to express our sincere gratitude for all the constructive comments on our manuscript. The comments and suggestions are very helpful for the improvement of our paper. In what follows, we present detailed comments in response to the individual points raised by the reviewer and elaborate on how the manuscript has been revised.

**(The line numbers referred in the response are those in the manuscript tracking changes.)**

**I. MAJOR COMMENTS**

**Comment 1:** *The authors construct a network using gridded VTEC data. Using the degree distribution and estimates of distance and clustering coefficient they conclude that the network has properties of small-world network but it is not a scale-free network. That the network is not a scale-free network is rather obvious given the degree distribution and the fact that there are no supernodes in the constructed network. Because of that I think that the fractal analysis (the box-counting approach) is not necessary (maybe it should be supplementary material).*

**Response:** Many thanks for the comments. As the reviewer suggested, we have removed the fractal analysis about the ionospheric network.

The revisions are shown in lines 11-13 in page 1, line 9 in page 3 and lines 9-11 in page 11.

**Comment 2:** *The authors state in the abstract and conclusion that "The analysis of small-world-ness indicates that the ionospheric network possesses the small-world property, which can make the ionosphere stable and efficient in the propagation of the dynamic processes. The fractal analysis shows that the ionospheric network is not self-similar in the current temporal and spatial resolution, indicating the complexity of the spatial variation for a long time in the ionosphere". I think this is very vague. Can the authors explain what exactly "stable" and "efficient" mean vis-á-vis the dynamics in the ionosphere? Does this result provide any new insights? I would like to see something more than the network a small-world network, therefore is stable and efficient.*

**Response:** Many thanks for the comments. We are sorry for the unclear explanation about "stable" and "efficient".

As for a complex network, the concept of "stable" is defined as the high capability of the dynamics in the network against the disturbance attacks. In other words, the topology structure of the stable network cannot be easily destroyed and the dynamics can still be propagated throughout the network, even when some edges are removed by the disturbance attacks. "Efficient" is defined as the ability about the rapid and easy propagation of dynamics in the network.

As was defined by Watts and Strogatz (1998), the small world network possesses a small average shortest path length (compared to the regular network) and a large clustering coefficient (compared to the random network). When the number of edges per node is high, networks would have a high clustering coefficient. In this case accidental removal of some edges does not break the network into nonconnected parts; the network is stable. On the other hand, the characteristic path length $L$ is the average shortest path length between two nodes. A small average shortest path length $L$ means faraway nodes can be connected as easily as nearby nodes. The smaller $L$, the easier the propagation is in the network. Within the networks with small $L$, the propagation of dynamics is efficient. Thus, small-world networks exhibit efficient dynamic propagation and at the same time are stable.

As is shown by the results in the subsection 3.3, the ionospheric network is small-world with a small average shortest path length and a large clustering coefficient. Thus, the ionospheric network exhibits properties of stable networks and of networks where dynamic processes are transferred efficiently. For example, the solar flare may create a disturbance in the ionosphere at high latitudes. However, the small world property of the ionospheric network allows the system to respond quickly and coherently to the anomalies introduced into the system. This dynamics propagation diffuses local anomalies thereby reducing the possibility of prolonged local extremes and providing greater stability for the global ionosphere system. Thus, chances of major ionospheric shifts are reduced. The above theory and its application to the ionosphere data suggest that the ionosphere system may be inherently stable and efficient in transferring dynamics.

The revisions are shown in lines 14-17 in page 8 and lines 19-32 in page 9.

**Comment 3:** *Also, the authors state on the bottom of page 2 that "Meanwhile, based on the causal relationship in the ionosphere, we can make a more precise prediction of the VTEC utilizing the observations obtained at the connected GIM cells in the network". I dont see anywhere in the paper anything about prediction.*

Response: Many thanks for the comments.

Here we mainly want to illustrate the potential function of the ionospheric network. Because the ionospheric network indicates the causal interactions among the GIM cells, we believe involving the information about the causal interactions is helpful to improve the prediction of the VTEC utilizing the observations obtained at the connected GIM cells in the network. The precise prediction of the VTEC based on the ionospheric networks is our current research topic.

As the reviewer commented, there are no details about prediction in the manuscript. Therefore, we think it is better to remove this part.

The revisions are shown in lines 34-35 in page 2 and lines 1-2 in page 3.

**Complex networks description of the ionosphere**

Shikun Lu[1,2], Hao Zhang[1], Xihai Li[2], Yihong Li[2], Chao Niu[2], Xiaoyun Yang[2], and Daizhi Liu[2]

[1]Department of Electronic and Engineering, Tsinghua University, Beijing, China.
[2]Xi'an Research Institute of Hi-Tech, Xi'an, China.

*Correspondence to:* Hao Zhang (haozhang@mail.tsinghua.edu.cn)

**Abstract.** Complex networks have emerged as an essential approach of geoscience to generate novel insights into nature of geophysical systems. To investigate the dynamic processes in the ionosphere, a directed complex network is constructed based on the probabilistic graph by the Vertical Total Electron Content (VTEC) in 2012. The results of the power-law hypothesis testing show that both the out-degree and in-degree distribution of the ionospheric network are not scale-free. Thus, the distribution of the interactions in the ionosphere is homogenous. None of the geospatial positions plays an eminently important role in the propagation of the dynamic ionospheric processes. The spatial analysis of the ionospheric network shows that the inter-connections principally exist between adjacent geographical locations, indicating that the propagation of the dynamic processes primarily depends on the geospatial distance in the ionosphere. Moreover, the joint distribution of the edge distances with respect to longitude and latitude directions shows that the dynamic processes travel further along the longitude than along the latitude in the ionosphere. The analysis of small-world-ness indicates that the ionospheric network possesses the small-world property, which can make the ionosphere stable and efficient in the propagation of dynamic processes.

*Copyright statement.* All authors have approved the manuscript for submission. The content of the manuscript has not been published or submitted elsewhere.

[revised manuscript text omitted]

**As for a complex network, the concept of "stable" is defined as the high capability of the dynamics in the network against the disturbance attacks. In other words, the topology structure of the stable network cannot be easily destroyed and the dynamics can still be propagated throughout the network, even when some edges are removed by the disturbance attacks. "Efficient" is defined as the ability about the rapid and easy propagation of dynamics in the network.** In this subsection, we explore the small-world structure of the ionospheric network to examine the stability and efficiency of the ionosphere which is regarded as a dynamical system.

Lying between the completely random and completely regular network, the small-world network is a type of graph in which any given node is likely to reach every other node by a small number of steps compared with the total number of network nodes (Gallos et al., 2007). The 'six degrees of separation' in social networks is one of the most famous examples. Watts and Strogatz (1998) initially found that some networks can be highly clustered, like regular lattices, yet have small characteristic path lengths, like random graphs. Networks of such nature is called small-world networks. To investigate the small-world structure of the ionospheric network, the original network has to be reduced to an undirected graph (Abe and Suzuki, 2006, 2009). Furthermore, to mathematically describe the small-world property, two critical parameters are often selected, which are the average clustering coefficient $C$ and the average shortest path length $L$. Their definitions are shown in Equation (2), (3) and (4).

$$C_i = \frac{2\Delta_i}{k_i(k_i - 1)}, \tag{2}$$

$$C = \frac{1}{N} \sum_{i=1}^{N} C_i, \tag{3}$$

$$L = \frac{2}{N(N-1)} \sum_{i \geq j} d_{ij}. \tag{4}$$

Here, $C_i$ is the local clustering coefficient of node $i$. $k_i$ is the degree of node $i$ and $\Delta_i$ denotes the number of edges between the neighbors of node $i$ with node $i$ itself being excluded. The global clustering coefficient $C$ is defined as the average of all local clustering coefficients $C_i$. $N$ is the number of nodes and $d_{ij}$ denotes the length of the shortest path between the nodes $i$ and $j$. $d_{ij}$ is calculated by Dijkstra algorithm (Newman, 2010). Thus, $C$ describes the local connections in the ionospheric networks, while $L$ characterizes a network's connectivity structure globally (Zerenner et al., 2014).

To quantitatively define a small-world network, values for the network properties must be compared with those values acquired from the equivalent random networks, which have the same degree with the given network on average. A measurement of 'small-world-ness' is proposed as follows (Humphries and Gurney, 2008; Humphries et al., 2011):

$$\sigma = \frac{C/C_r}{L/L_r}. \tag{5}$$

Here, $C$ and $L$ are the average clustering coefficient and the average shortest path length of the given network, while $C_r$ and $L_r$ are those of the equivalent random network. If the given network fulfills the conditions, $\sigma > 1$ and $C/C_r > 1$, it meets the small-world criteria. To reduce the impact of randomness during the analysis of the ionospheric network, the results shown in Figure 4 are calculated by 150 random networks.

From Figure 4 (a) and (c), we can find that the results all satisfy $\sigma > 1$ and $C/C_r > 1$. Shown in Figure 4 (b) and (d), the frequencies are approximately Gaussian, and the standard deviations are 0.028 and 0.035. Such small standard deviations indicate the results are close to the real values (the averages) 6.64 and 8.08. Therefore, the ionospheric network behaves as a small-world graph. The propagation of the dynamic processes in the ionosphere presents small-world property. **As was defined by Watts and Strogatz (1998), the small world network possesses a small average shortest path length (compared to the regular network) and a large clustering coefficient (compared to the random network). When the number of edges per node is high, networks would have a high clustering coefficient. In this case, accidental removal of some edges does not break the network into nonconnected parts; the network is stable. On the other hand, a small average shortest path length $L$ means faraway nodes can be connected as easily as nearby nodes. The smaller $L$, the easier the propagation is in the network. Within the networks with small $L$, the propagation of dynamics is efficient. Thus, small-world networks are stable and efficient to react to the abrupt variations (Tsonis et al., 2007).**

**As is shown by the results above, the ionospheric network is small-world with a small average shortest path length and a large clustering coefficient. Thus, the ionospheric network exhibits properties of stable networks and of networks where dynamic processes are transferred efficiently. For example, the solar flare may create a disturbance in the ionosphere at high latitudes. However, the small world property of the ionospheric network allows the system to respond quickly and coherently to the anomalies introduced into the system. This dynamic propagation diffuses local anomalies thereby reducing the possibility of prolonged local extremes and providing greater stability for the global ionosphere**

[revised manuscript text omitted]